# Protecting Copyrighted Material with Unique Identifiers in Large Language Model Training

## Abstract

A primary concern regarding training large language models (LLMs) is whether they abuse copyrighted online text. With the increasing training data scale and the prevalence of LLMs in daily lives, two problems arise: 1) false positive membership inference results misled by similar examples; 2) membership inference methods are usually too complex for general users to understand and use. To address these issues, we propose an alternative *insert-and-detect* methodology, advocating that web users and content platforms employ **unique identifiers** for reliable and independent membership inference. Users and platforms can create their identifiers, embed them in copyrighted text, and independently detect them in future LLMs. As an initial demonstration, we introduce **ghost sentences** and a user-friendly last-$k$ words test, allowing general users to chat with LLMs for membership inference. Ghost sentences consist primarily of unique passphrases of random natural words, which can come with customized elements to bypass possible filter rules. The last-$k$ words test requires a significant repetition time of ghost sentences ($\geq 10$). For cases with fewer repetitions, we designed an extra perplexity test, as LLMs exhibit high perplexity when encountering unnatural passphrases. We also conduct a comprehensive study on the memorization and membership inference of ghost sentences, examining factors such as training data scales, model sizes, repetition times, insertion positions, wordlist of passphrases, alignment, *etc*. Our study shows the possibility of applying ghost sentences in real scenarios and providing instructions for the potential application.

## 1 Introduction

Large language models (LLMs) are pre-trained on vast amounts of data sourced from the Internet, while the providers of commercial LLMs like ChatGPT, Bard, and Claude do not disclose the details of the training data. This raises concerns that LLMs may be trained with copyrighted material without permission from the creators (Karamolegkou et al., 2023; Henderson et al., 2023; Li et al., 2024). Some efforts have been made to determine whether a specific example is included in the training data (Mattern et al., 2023; Meeus et al., 2024; Shi et al., 2024; Li et al., 2024). However, the false positive membership inference results caused by similar examples are common (Duan et al., 2024). Service providers might argue that detection results could be confused by similar examples in massive data rather than the exact copyrighted content (OpenAI, 2019). Additionally, these membership inference methods are often too complex for general users without coding experience or expert knowledge. This complexity could lead to centralized detection services, which reduces transparency and raises concerns about trustworthiness.

For transparent and reliable protection of copyrighted material[1], we propose an alternative *insert-and-detect* methodology for general web users and content platforms (*e.g.*, Quora, Medium, Reddit, GitHub). We advocate that web users and content platforms insert **unique identifiers** into copyrighted content. These identifiers help address the issue of false positives caused by similar examples (OpenAI, 2019; Duan et al., 2024), providing definitive evidence for copyright protection. The process should be transparent, allowing users and content platforms to create unique identifiers, embed them in online copyrighted material, and perform detection independently.

---

[1] Any creative, intellectual, or artistic text presented on the Internet, such as poems, blogs, fiction, and code.

Figure 1: **Insertion and test of ghost sentences.** A ghost sentence primarily consists of a unique passphrase, with customizable elements like a prefix added to bypass potential filters. Given an LLM, users can conduct a last-$k$ words test by interacting with the LLM for reliable membership inference. Alternatively, users can perform a perplexity test if prediction scores are available.

To demonstrate the concept, we introduce **_ghost sentences_** as a primitive implementation of unique identifiers, as well as a user-friendly last-$k$ words test for their membership inference. A ghost sentence is distinctive because it primarily consists of a randomly generated diceware passphrase (Reinhold, 1995). As shown in Figure 1, users or content platforms can insert a ghost sentence, along with customized elements, into various online documents. When the repetition of ghost sentences increases, LLMs are likely to achieve verbatim memorization (Carlini et al., 2019; 2021; Ishihara, 2023; Karamolegkou et al., 2023) of the passphrases in ghost sentences. In this case, users can prompt LLMs to complete the last $k$ words of a ghost sentence, using the preceding context, as shown in Figure Figure 1. For example, the last-$k$ word test can be performed on ChatGPT using content from popular books, as demonstrated in Figure 5. Due to the randomness of passphrases, it is statistically guaranteed that if an LLM can complete the last $k \geq 1$ words, it must have been trained with the ghost sentence. In experiments with an OpenLLaMA-3B (Geng & Liu, 2023) model, 11 out 16 users successfully identify their data from the LLM generation. These 16 users have 24 examples with ghost sentences on average and contribute 383 examples to a total of 1.8M training documents. Ghost sentences account for only 0.0017% of all training tokens.

The last $k$ words test is user-friendly but requires a non-trivial repetition time of ghost sentences ($\geq 10$). Following previous membership inference methods based on loss, entropy, or probability of predictions (Yeom et al., 2018; Carlini et al., 2021; Shi et al., 2024), we design an alternative perplexity test for less frequently repeated ghost sentences. An LLM trained with natural languages should exhibit high perplexity for the passphrase in a ghost sentence, as it consists of random words. During the perplexity test, users can generate a new set of ghost sentences, obtain the perplexity distribution, and use the distribution to perform a hypothesis test for membership inference. For a LLaMA-13B model (Touvron et al., 2023a), a perplexity test for 30 ghost sentences, with an average repetition of 7 in 148K examples, achieves a 0.891 ROC AUC.

We also comprehensively study different factors influencing the memorization and membership inference results of ghost sentences. A few key observations are as follows: 1) The memorization of ghost sentences is jointly decided by their quantity and average repetition. Ghost sentences with a word length $\geq 8$, an average repetition $\geq 5$, and a proportion $\geq 0.0016\%$ of training tokens are recommended. 2) It is better to insert ghost sentences in the latter half of a document. 3) A curated wordlist for the generation of passphrases is necessary. We suggest using a well-maintained wordlist from the Electronic Frontier Foundation. 4) Further model alignment (Ouyang et al., 2022; Rafailov et al., 2023) will not affect the memorization of ghost sentences. 5) The larger the model, the smaller the repetition times for memorization. This is consistent with previous works (Carlini et al., 2023). Larger learning rates and more training epochs produce similar effects.

A single pattern of unique identifiers is insufficient, as it may eventually be filtered out, despite the significant cost of filtering hidden sentences from terabytes or even petabytes of data. As LLMs become increasingly popular in daily lives, there is a growing need for diverse unique identifiers and user-friendly test methods. Different copyright identifiers are not mutually exclusive and can work together to make filtering intractable. Wei et al. (2024) adopt random characters, which also qualify as unique identifiers. Nevertheless, relying solely on long random characters risks filter through measures like regular expression matching and semantic checking. Additionally, random characters, such as auto-generated metadata, are prevalent in large-scale datasets (Elazar et al., 2024), which can lead to false detection issues (Duan et al., 2024). They also lack a user-friendly membership inference method for general users. We hope ghost sentences can serve as a starting point for creating diverse unique identifiers and user-friendly membership inference methods.

## 2 RELATED WORKS

**Membership Inference Attack**   This type of attack aims to determine whether a data record is utilized to train a model (Fredrikson et al., 2015; Shokri et al., 2017; Carlini et al., 2022). Typically, membership inference attacks (MIA) involve observing and manipulating confidence scores or loss of the model (Yeom et al., 2018; Song & Shmatikov, 2019; Mattern et al., 2023), as well as training an attack model (Shokri et al., 2017; Hisamoto et al., 2020). Duan *et al.* (Duan et al., 2024) conduct a large-scale evaluation of MIAs over a suite of LLMs trained on the Pile (Gao et al., 2020) dataset and find MIAs barely outperform random guessing. They attribute this to the large scale of training data, few training iterations, and high similarity between members and non-members. Shi *et al.* (Shi et al., 2024) utilize wiki data created after LLMs training to distinguish the members and non-members. Nevertheless, the concern that similar examples in the large-scale training data may lead to ambiguous inference results remains.

**Machine-Generated Text Detection**   Text watermark (Kirchenbauer et al., 2023; Gu et al., 2024; Liu et al., 2024; 2023; Ding et al., 2024) aims to embed signals into machine-generated text that are invisible to humans but algorithmically detectable. Generally, LLMs are required not to generate tokens from a red list. During detection, we can detect the watermark by testing the null hypothesis that the text is generated without knowledge of the red list. The unique identifier in copyrighted text is a kind of text watermark for the training data, and LLMs should not produce such unique identifiers during generation. A few other methods (Mitchell et al., 2023; Bao et al., 2024; Mireshghallah et al., 2024) try to detect machine-generated text without modifying the generation content. They are mainly based on the assumption that the patterns of log probabilities of human-written and machine-generate text have distinguishable discrepancies.

**Training Data Extraction Attack**   The substantial number of neurons in LLMs enables them to memorize and output part of the training data verbatim (Carlini et al., 2023; Ishihara, 2023; Zhang et al., 2023). Adversaries exploit this capability of LLMs to extract training data from pre-trained LLMs (Carlini et al., 2021; Nasr et al., 2023; Lee et al., 2023; Kudugunta et al., 2023). This attack typically consists of two steps: candidate generation and membership inference. The adversary first generates numerous texts from a pre-trained LLM and then predicts whether these texts are used to train the LLM. Carlini *et al.* (Carlini et al., 2023) quantify the memorization capacity of LLMs, discovering that memorization grows with the model capacity and the duplicated times of training examples. Specifically, within a model family, larger models memorize $2 - 5\times$ more than smaller models, and repeated strings are memorized more. Karamolegkou *et al.* (Karamolegkou et al., 2023) demonstrate that LLMs can achieve verbatim memorization of literary works and educational material. We also provide an similar example in Figure 5 in §App. C.

## 3 METHODOLOGY

### 3.1 PRELIMINARIES

Recent LLMs typically learn through language modeling in an auto-regressive manner (Bengio et al., 2003; Radford et al., 2019; Brown et al., 2020). For a set of examples $\mathcal{X} = \{x_1, x_2, \ldots, x_n\}$, each consisting of variable length sequences of symbols $x = \{s_1, s_2, ..., s_l\}$, where $l$ is the length of example $x$. During training, LLMs are optimized to maximize the joint probability of $x$: $p(x) = \prod_{i=1}^{l} p(s_i|s_1, \ldots, s_{i-1})$.

We assume there is a subset of examples $\mathcal{G} \subseteq \mathcal{X}$ from $m$ users that contain unique identifiers (ghost sentences in this work). Each user owns a set of examples $\mathcal{G}_i$ and $\mathcal{G} = \bigcup_{i=1}^{m} \mathcal{G}_i$. Without loss of generality, we assume there is only a unique ghost sentence in $\mathcal{G}_i$, which is repeated for $|\mathcal{G}_i|$ times. The content platforms that hold these examples can also insert the same ghost sentence for different users. The average repetition times of ghost sentences is $\mu = |\mathcal{G}|/m$. In subset $\mathcal{G}$, an example with a ghost sentence $g = \{w_1, w_2, \ldots, w_q\}$ becomes $(s_1, \ldots, s_j, w_1, \ldots, w_q, s_{j+1}, \ldots, s_l)$, where $q$ is the length of $g$ and $j$ is the insertion position. The joint probability of the ghost sentence is maximized during training: $p(g) = \prod_{i=1}^{q} p(w_i|s_1, \ldots, s_j, w_1, \ldots, w_{i-1})$.

**Creation of Ghost Sentences**   The main part of a ghost sentence is a diceware passphrase (Reinhold, 1995). Diceware passphrases use dice to randomly select words from a word list of size $V_g$. $V_g$ is generally equal to $6^5 = 7776$, which corresponds to rolling a six-sided dice 5 times. For a

diceware passphrase with length $q$, there are $7776^q$ possibilities, ensuring the uniqueness of a ghost sentence when $q \geq 4$, which is much larger than the number of indexed webpages estimated by worldwidewebsize.com (at least 2.37 billion indexed pages, October, 2024). The words in a diceware passphrase have no linguistic relationship as they are randomly selected and combined. Users can customize ghost sentences by add prefixes to passphrases as shown in Figure 1. It is recommended to use passphrases with more than 8 words and insert ghost sentences in the latter half of a document. We provide a few examples of ghost sentences in §App. G.

**Statistics of Users on Reddit**  In §App. D, we provide the statistics of users in Webis-TLDR-17 (Völske et al., 2017), a subset of Reddit data contains 3.8M examples from 1.4M users. The distribution of the number of document per user is long-tailed. Users with more than 4 and 9 examples contribute 41% and 22% of all data, respectively. These users can insert ghost sentences by themselves, other users contribute about 60% examples may need assistance from the content platform.

**Null Hypothesis**  We detect ghost sentences by testing the following null hypothesis,

$$H_0\text{: The LLM is trained with no knowledge of ghost sentences.} \tag{1}$$

## 3.2 LAST-$k$ WORDS TEST

During inference or generation, users can request the LLM to output the last $k$ words of a ghost sentence $g$ given their preceding context $c$ as input prompt:

$$w^\star_{l-k+1} = \texttt{Gen}(c, w_1, \ldots, w_{l-k}). \tag{2}$$

Here, $l$ is the total length, $\texttt{Gen}(\cdot)$ represents the generation function, and $w^\star_i$ is the predicted word.

If the null hypothesis is true, at each step, the probability of the LLM generates a correct word corresponds to that in the passphrase is $1/V^\star$, where $V_g \leq V^\star$ and $V_g$ is the vocabulary size of random words. Suppose we are generating a passphrase of length $q$, the number of correct words at all steps, $n_g$, has an expected value $q/V^\star$ and a variance $q(V^\star - 1)/(V^\star)^2$. We can perform a *one proportion z-test* to evaluate the null hypothesis, and the $z$-score for the test is:

$$z = \frac{n_g V^\star - q}{\sqrt{q(V^\star - 1)}}. \tag{3}$$

Suppose the length of passphrase $q = 10$ and $V^\star = 7,776$, with $n_g = 1$. This results in a z-score of $27.85 \gg 2.58$; the latter is at a significant level of 0.01. In this case, we reject the null hypothesis, and the probability of a false positive is nearly 0. In practice, as ghost sentences in the training data increase, $1/V^\star$ also increases, and a large $n_g$ may be required for the test. When $n_g = 2$, the test can reject the null hypothesis even if $1/V^\star = 1/25$ at a significant level 0.01. A probability $1/25$ is clearly not normal for generating random words. Our analysis for ghost sentence detection is similar to that for detecting text watermark (Kirchenbauer et al., 2023).

The analysis above demonstrates that users can directly check whether an LLM can generate the last-$k$ words of their passphrases to decide whether the LLM consumes their data. $k = 1$ or $k = 2$ can already guarantee the robustness of test results. To understand how many repetition times for ghost sentences are required for the last-$k$ words test, we define two quantitative metrics: *document identification accuracy* ($\texttt{D-Acc}$) and *user identification accuracy* ($\texttt{U-Acc}$):

$$\texttt{D-Acc-}k_\mathcal{G} = \frac{1}{|\mathcal{G}|} \sum_{g \in \mathcal{G}} \prod_{i=1}^{k} \mathbf{1}\{w^\star_{l-i+1} = w_{l-i+1}\}, \tag{4}$$

$$\texttt{U-Acc-}k = \frac{1}{m} \sum_{i}^{m} \mathbf{1}\{\texttt{D-Acc-}k_{\mathcal{G}_i} > 0\}, \tag{5}$$

where $\mathbf{1}\{\cdot\}$ equals 1 if the inner condition is true, 0 otherwise. Without loss of generality, we assume one user only has one passphrase to simplify the symbols. $\texttt{D-Acc-}k_\mathcal{G}$ assesses the memorization successful rate of the last $k$ words for the document set $\mathcal{G}$, and $\texttt{U-Acc-}k$ evaluates the accuracy for user identities. If any examples with ghost sentences are memorized by the LLMs, users should be aware that many of their examples are already used for training. Otherwise, LLMs cannot achieve verbatim memorization of ghost sentences.

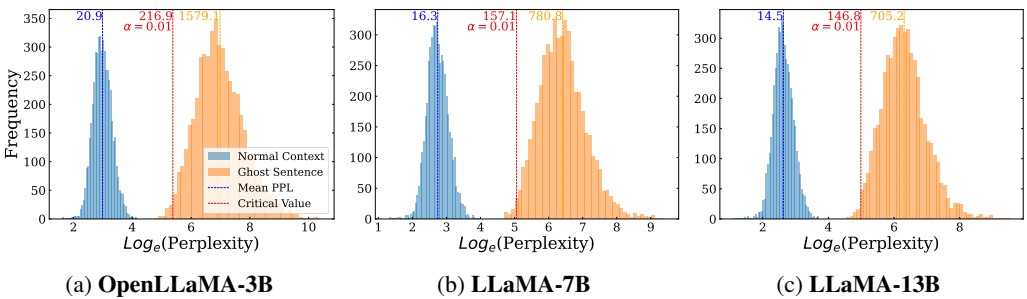

(a) **OpenLLaMA-3B**  (b) **LLaMA-7B**  (c) **LLaMA-13B**

Figure 2: **Perplexity Discrepancy between normal context and ghost sentences**. We randomly generate 5,000 ghost sentences and insert them into 5,000 examples from Webis-TLDR-17.

### 3.3 PERPLEXITY TEST

The last $k$ words test is user-friendly but requires a significant repetition time ($> 10$) to achieve verbatim memorization of ghost sentences. Inspired by previous membership inference methods based on loss, entropy, or probability of predictions (Yeom et al., 2018; Carlini et al., 2021; Shi et al., 2024), we design a perplexity test for less repeated ghost sentences. The perplexity of a ghost sentence $g = \{w_1, w_2, \ldots, w_q\}$ given context $c = (s_1, \ldots, s_j)$ is:

$$\mathrm{PPL}(g) = \exp\Big\{-\frac{1}{q}\sum_{i=1}^{q} \log p(w_i|c, w_{<i})\Big\}. \tag{6}$$

For simplicity, we only consider the perplexity of passphrases, excluding customized elements. Passphrases are combinations of random words. If the null hypothesis is true, the LLM is basically doing random guessing given a vocabulary $V$, and the value of $\mathrm{PPL}(g)$ should be high.

Figure 2 presents the perplexity discrepancy between normal context ($\mathrm{PPL}(c)$) and ghost sentences ($\mathrm{PPL}(g)$ given $c$). On average, the perplexity of ghost sentences are much higher than that of natural language. Given an LLM, a ghost sentence $g$, and a context $c$, we can use the empirical perplexity distribution of ghost sentences (unseen by the LLM) to perform a hypothesis test. If $\mathrm{PPL}(g)$ is smaller than the critical value at a certain significant level, we will reject the null hypothesis $H_0$. For example, if $\mathrm{PPL}(g) < 157$ for a LLaMA-7B model in Figure 2, we will reject $H_0$ and the probability of a false positive is less than 1%. The perplexity test requires one ghost sentence to be repeated a few times in the training data of LLMs. For a LLaMA-13B model fine-tuned on 148K examples with 30 ghost sentences repeat 5 times on average, a perplexity test can achieve 0.393 recall with a significant level 0.05 after 1 epoch fine-tuning. The recall increases to 0.671 if the average repetition becomes 7.

### 3.4 LIMITATIONS

As a primitive design of unique identifiers for demonstration, ghost sentences offer both advantages and limitations. They are transparent, user-friendly, and statistically trustworthy. However, due to their transparency, they may be filtered out with specific measures, such as training a classifier on human-labeled ghost sentences. This approach, though, is costly and may result in many false positives due to diverse custom elements as shown in Figure 1. Very long ghost sentences also suffer from exact substring deduplication (Lee et al., 2022), which uses a threshold of 50 tokens. Therefore, we recommend using a passphrase of around 10 words, which is 22 tokens on average for BPE tokenizer (Sennrich et al., 2015). Actually, service providers do not adopt a strict deduplication process, as verbatim memorization of popular books can still be found (Karamolegkou et al., 2023) (or Figure 5). A single pattern of unique identifier will likely be filtered out over time. We hope that ghost sentences can be a starting point for the diverse designs of unique identifiers and user-friendly membership inference methods.

## 4 EXPERIMENTS

### 4.1 EXPERIMENTAL DETAIL

In this work, we consider inserting ghost sentences at the pre-training stage and instruction tuning (Wang et al., 2023; Taori et al., 2023) stage. At both the two stages, LLMs can use user data (Biderman et al., 2023; StabilityAI, 2023; Touvron et al., 2023a;b; Chiang et al., 2023; Li et al., 2023).

**Models**   For instruction tuning, we adopt the LLaMA serires (Touvron et al., 2023a), including OpenLLaMA-3B (Geng & Liu, 2023), LLaMA-7B, and LLaMA-13B. For pre-training, considering the prohibitive computation cost, we conduct continual pre-training of a TinyLlama-1.1B model at 50K steps(TinyLlama/TinyLlama-1.1B-step-50K-105b), 3.49% of its total 1431K training steps. The context length of all models is restricted to 512. The batch size for instruction tuning is 128 examples following previous works (Taori et al., 2023; Li et al., 2023). We maintain the pre-training batch size the same as TinyLlama-1.1B — 1024 examples. A large batch size is achieved with gradient accumulation on 4 NVIDIA RTX A6000 GPUs.

**Training Epochs and Learning Rate**   *All models are only trained for **1 epoch**.* Actually, training epochs of LLaMA range from $0.64 \sim 2.45$ for different data. As for the learning rate, we keep consistent with LLaMA or TinyLlama with a linear scaling strategy. Specifically, our learning rate is equal to $\frac{\text{our batch size}}{\text{original batch size}} \times$ original learning rate. LLaMA-7B uses a batch of 4M tokens with a 3e-4 learning rate, so our learning rate for instruction tuning is 3e-4 $\times \frac{128 \times 512}{4 \times 2^{20}} \approx$ 4.6e-6. TinyLlama uses learning ate 4e-4, batch size 1024, and context length 2048, so our learning rate for continuing pre-training is 1e-4. By default, the optimizer is AdamW (Loshchilov & Hutter, 2017) with a cosine learning rate schedule. All models are trained with mixed precision and utilize FlashAttention (Dao et al., 2022; Dao, 2023) to increase throughput.

**Dataset**   Webis-TLDR-17 (Völske et al., 2017) contains 3.7M examples with word lengths under 4096. Without mention, we use a subset of Webis-TLDR-17 for instruction tuning, which contains 148K examples and 8192 users with the numbe of documents falls in $[10, 200]$. We term this subset as *Webis-148K* for convenient. For instruction tuning on Webis-148K, LLMs are required to finish a continue writing task using the instruction `"Continue writing the given content"`. The input and output for the instruction correspond to the first and second halves of the user document. For continuing pre-training, we also utilize the LaMini-Instruction (Wu et al., 2023) and OpenOrca (Longpre et al., 2023; Mukherjee et al., 2023; Lian et al., 2023) datasets, which contain 2.6M and 3.5M examples, respectively. Plus the Webis-TLDR-17 dataset, the number of pre-training examples is 9.8M. All data are shuffled during training.

**Evaluation and Metrics**   For perplexity test, we calculate the detection accuracy, *i.e.*, the ratio of correctly detected examples among all samples with ghost sentences after performing hypothesis test. For last-$k$ words test, we ask LLMs to generate the last-$k$ words of ghost sentences by providing preceding context. A beam search with width 5 is used for generation. `D-Acc-`$k$ and `U-Acc-`$k$ are calculated with $k = 1$ and $k = 2$.

### 4.2 PERPLEXITY TEST

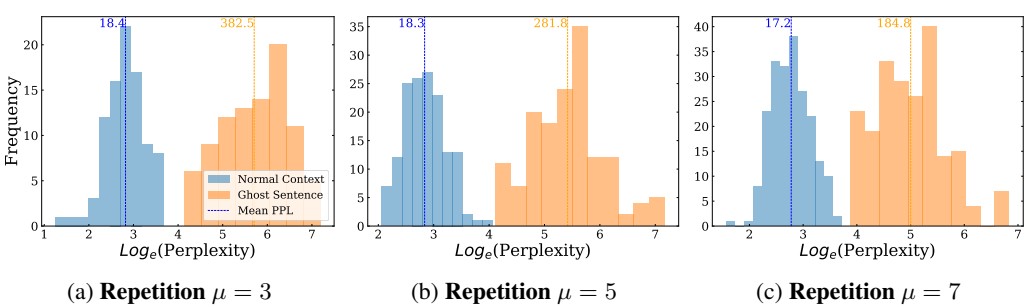

|   (a) **Repetition** $\mu = 3$   |   (b) **Repetition** $\mu = 5$   |   (c) **Repetition** $\mu = 7$   |

Figure 3: **Perplexity of a fine-tuned LLaMA-7B model**. 30 unique ghost sentences in Webis-148K. As the repetition times increase, the perplexity of ghost sentences (`PPL`$(g)$ given $c$) decreases.

To figure out the average repetition $\mu$ of ghost sentences for the perplexity test, we randomly generate 30 different ghost sentences with a word length 10. Then we randomly select $30 \times \mu$ examples from Webis-148K and insert ghost sentences at the end of these examples.

Table 1 presents the ROC AUC and recall of a perplexity test after fine-tuning LLaMA models on Webis-148K. During the test, we sample the same number of non-member examples from Webis-TLDR-17 and insert newly generated ghost sentences into them. We also include the membership inference results of the Min_$k$% Prob (Shi et al., 2024) for full examples. The recall corresponds to a significant level 0.05, and we choose a critical value like

Table 1: **AUC and recall of the perplexity test**. prop.(%) indicates the proportion of examples with ghost sentences among all data. The critical value is 200.0 for recall.

| $\mu$ | prop.(%) | LLaMA-7B | | LLaMA-13B | |
|---|---|---|---|---|---|
| | | AUC | Recall | AUC | Recall |
| 1 | 0.02 | 0.542 | 0.033 | 0.558 | 0.033 |
| 3 | 0.06 | 0.745 | 0.030 | 0.747 | 0.289 |
| 5 | 0.10 | 0.805 | 0.393 | 0.808 | 0.453 |
| 7 | 0.14 | 0.883 | 0.671 | 0.891 | 0.710 |
| 9 | 0.18 | **0.902** | **0.770** | **0.991** | **0.904** |
| Min_$k$% Prob (Shi et al., 2024) ($\mu = 9$, full example) | | | | | |
| Min_5% Prob | | 0.600 | 0.513 | 0.761 | 0.707 |
| Min_10% Prob | | 0.583 | 0.565 | 0.720 | 0.682 |

Figure 2 ($\sim$200). When the repetition $\mu \geq 5$, the perplexity test starts to provide a decent performance. Figure 3 displays the perplexity of the LLaMA-7B models fine-tuned with ghost sentences. With an increase in repetition times, we observe a dramatic decrease in the perplexity of ghost sentences. *For every two additional repetitions of ghost sentences, the average perplexity decreases by $\sim$100.* The perplexity of normal context is roughly the same after fine-tuning.

## 4.3 Last-$k$ Words Test

In this section, we will figure out the conditions under which LLMs can achieve verbatim memorization of ghost sentences for the last-$k$ words test. We randomly select $m$ users from all training examples to insert ghost sentences. Each user has a unique ghost sentence, and the average repetition times of ghost sentences is $\mu$. A few key observations:

- When $\mu \geq 10$, ghost sentences with a word length of $\sim$10 are likely to be memorized by an OpenLLaMA-3B model fine-tuned on Webis-148k. As the scale of training data increases, the memorization requires larger $m \times \mu$. In most cases, we observe that a proportion of ghost sentence tokens to all tokens $\geq 0.0016\%$ is necessary (§4.3.1).

- The success rate of memorization is jointly determined by $m$ and $\mu$. Notably, $\mu$ is more critical than $m$. A ghost sentence with a small $\mu$ can become memorable with an increase in the number of different ghost sentences $m$ (§4.3.1).

- It is better to insert ghost sentences in the latter half of a document. The insertion of ghost sentences will not affect the linguistic performance of LLMs (§4.3.2, §App. E).

- Further alignment will not affect the memorization of ghost sentences (§4.3.4, §4.3.5).

- Training data domains and the choices of wordlists for passphrase generation also impact the memorization of ghost sentences (§4.3.5).

- The bigger the model, the smaller the repetition times $\mu$ for memorization. This is consistent with Carlini et al. (2023). Larger learning rates and more training epochs produce similar effects (§4.3.3).

### 4.3.1 Number and Repetition Times

*The number of ghost sentences $m$ and average repetition time $\mu$ work together to make an LLM achieve effective memorization.* Table 2a illustrates the influence of different $m$ and $\mu$. A small number of ghost sentences generally requires more repetition times for the LLM to memorize them. However, a large number of ghost sentences $m$ with small repetition times $\mu$ cannot achieve memorization. For example, the LLM cannot remember any ghost sentences of 16 users with $\mu = 13$, while a single user with repetition time 51 can make the LLM remember his ghost sentence.

*As the data increase, $m$ and $\mu$ should also increase accordingly.* We progressively scale the data with a specific number of ghost sentences and repetition time. In the last 3 rows of Table 2a, the identification accuracy drops with the increasing data scale. For 16 sentences with 24 average repetition time in 1.8M training examples, they can achieve 68.75% user identification accuracy when

Table 2: **Fine-tuning an OpenLLaMA-3B model with ghost sentences**. **(a) #Docs** represents the number of training examples, **mid.** is the median of $\mu$, and **prop.**(%) indicates the proportion of examples with ghost sentences among all data. **(b)** $100\%$ for position denotes insertion at the end of the example, and $[25, 100]$ means random insertion in the $25\% \sim 100\%$ of the example length $l$. $m = 256, \mu = 17$, median $= 13.5$, and 148K examples.

(a) **different $m$ and $\mu$.**

| #Docs | $m$ | $\mu$ | mid. | prop.(%) | $k=1$ U-Acc | $k=1$ D-Acc | $k=2$ U-Acc | $k=2$ D-Acc |
|---|---|---|---|---|---|---|---|---|
| | 0 | 0 | 0.00 | 0.00 | 0.00 | 0.00 | 0.00 | 0.00 |
| | 256 | 17 | 13.5 | 2.99 | 92.58 | 91.01 | 84.77 | 84.66 |
| | 128 | 17 | 13.0 | 1.47 | 85.94 | 85.96 | 73.44 | 75.26 |
| | 64 | 17 | 13.0 | 0.74 | 56.25 | 64.62 | 48.44 | 57.56 |
| | 32 | 18 | 12.0 | 0.39 | 75.00 | 78.86 | 65.62 | 74.18 |
| 148K | 16 | 13 | 11.5 | 0.14 | 0.00 | 0.00 | 0.00 | 0.00 |
| | 16 | 21 | 16.5 | 0.22 | 62.50 | 64.85 | 50.00 | 55.15 |
| | 8 | 18 | 13.0 | 0.10 | 25.00 | 26.76 | 12.50 | 25.35 |
| | 8 | 31 | 25.5 | 0.16 | 100.0 | 94.29 | 100.0 | 90.61 |
| | 4 | 32 | 32.5 | 0.09 | 50.00 | 37.21 | 0.00 | 0.00 |
| | 2 | 48 | 47.5 | 0.06 | 100.0 | 98.95 | 100.0 | 98.95 |
| | 1 | 45 | 45.0 | **0.03** | 100.0 | 73.33 | 100.0 | 35.56 |
| | 1 | 51 | 51.0 | 0.03 | 100.0 | 98.04 | 100.0 | 98.04 |
| 148K | | | | 0.26 | 100.00 | 92.69 | 87.50 | 80.68 |
| 592K | 16 | 24 | 20.5 | 0.07 | 93.75 | 93.73 | 87.50 | 89.30 |
| 1.8M | | | | **0.02** | 68.75 | 67.89 | 43.75 | 42.82 |

(b) **sentence length and insertion position.**

| Length | Position (%) | $k=1$ U-Acc | $k=1$ D-Acc | $k=2$ U-Acc | $k=2$ D-Acc |
|---|---|---|---|---|---|
| 6 | | 87.50 | 84.59 | 77.34 | 74.36 |
| 8 | | 84.38 | 82.81 | 75.39 | 74.54 |
| 10 | | 89.06 | 86.60 | 80.47 | 79.20 |
| 12 | 100 | 92.58 | 91.01 | 84.77 | 84.66 |
| 14 | | 83.59 | 83.98 | 75.00 | 76.42 |
| 16 | | 91.02 | 89.72 | 84.77 | 85.43 |
| 18 | | 84.77 | 86.04 | 77.73 | 80.64 |
| 20 | | 91.41 | **92.64** | **86.33** | **87.35** |
| 12 | 50 | 35.94 | 3.68 | 34.38 | 3.59 |
| 12 | 75 | 48.83 | 6.08 | 47.66 | 5.87 |
| 12 | 100 | 92.58 | **91.01** | 84.77 | **84.66** |
| 12 | [25, 100] | 88.28 | 39.33 | 80.08 | 36.77 |
| 12 | [50, 100] | **94.53** | 59.50 | **89.84** | 57.47 |
| 12 | [75, 100] | 91.02 | 75.40 | 87.05 | 72.67 |

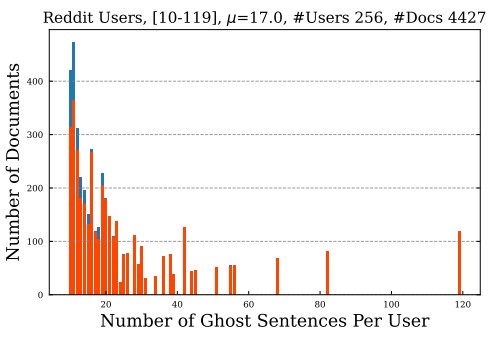

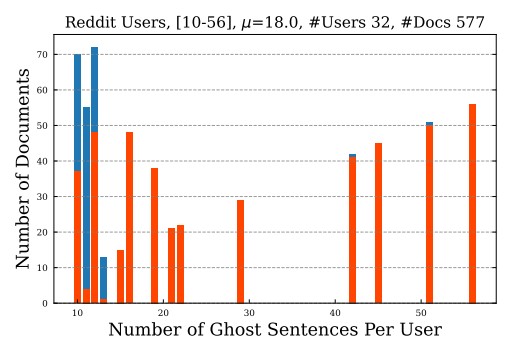

(a) $m = \mathbf{256}, \mu = \mathbf{17}$, median $= \mathbf{13.5}$.      (b) $m = \mathbf{32}, \mu = \mathbf{18}$, median $= \mathbf{12.0}$.

Figure 4: **D-Acc-1 with different repetition times of ghost sentences**. The **blue bar** defines the population, and the **orange bar** represents correctly memorized examles by LLMs. The total training data is 148K. Examples with ghost sentences in (**b**) are sampled from (**a**).

$k = 1$, namely, 11 of 16 users can get the correct last-1 word prediction. In this case, documents with ghost sentences only account for 0.02% of all 1.8M examples. The minimal average repetition time of these 16 ghost sentences is 16. For reference, Webis-TLDR-17 contains 17.8K users which have a document count exceeding 16. Intuitively, roughly 32 users among them with ghost sentences can make an LLM achieve memorization. This suggests that the practical application of ghost sentences is feasible. Content platforms can easily achieve such a goal.

*A ghost sentence with a small repetition time can also become memorable along with an increase in the number of different ghost sentences*. Figure 4 presents the D-Acc-1 with different repetition times of ghost sentences. In Figure 4a, when the number of documents with ghost sentences is large, ghost sentences with $\mu = 10$ or 11 can achieve $\sim 75\%$ D-Acc-1. Nevertheless, the D-Acc-1 dramatically decreases in Figure 4b, where the number of documents are only $\sim 25\%$ (577) of that in Figure 4a (4427). This is good news for users with a relatively low document count.

### 4.3.2 LENGTH AND INSERTION POSITION

*Longer ghost sentences are generally easier to memorize for the LLM*. In Table 2b, we gradually increase the length of the ghost sentences, and longer ghost sentences are more likely to get higher user and document identification accuracy. The reason is quite straightforward: as the length in-

Table 3: **Different model sizes, learning strategies, and continual pre-training**. **(a)** Training data is Webis-148K with ghost sentences, $m = 256, \mu = 17, \text{median} = 13.5$. ♠ means $m = 256, \mu = 29, \text{median} = 22.0$. **(b) mid.** is the median of repetition times, and **prop.**(%) is the proportion of examples with ghost sentences in all data. The length of ghost sentences is 12.

(a) **model sizes, learning rate, and epochs.**

| Params | lr | Epochs | k = 1 U-Acc | k = 1 D-Acc | k = 2 U-Acc | k = 2 D-Acc |
|---|---|---|---|---|---|---|
| 3B | 3.6e-6 | 1 | 67.52 | 67.58 | 54.80 | 51.56 |
| | 4.6e-6 | | 92.58 | 91.01 | 84.77 | 84.66 |
| | 5.6e-6 | | **96.09** | **98.05** | **92.73** | **93.36** |
| 3B | 3.6e-6 | **2** | **100.0** | **100.0** | **100.0** | **99.98** |
| 1.1B | 4.6e-6 | | 0.0 | 0.0 | 0.0 | 0.0 |
| ♠1.1B | 4.6e-6 | 1 | 85.16 | 84.92 | 77.96 | 75.00 |
| 3B | 4.6e-6 | | 92.58 | 91.01 | 84.77 | 84.66 |
| 7B | 4.6e-6 | | **98.05** | **98.03** | **97.27** | **97.40** |

(b) **continuing pr-training of TinyLlama-1.1B**.

| #Docs | $m$ | $\mu$ | mid. | prop.(%) | k = 1 U-Acc | k = 1 D-Acc | k = 2 U-Acc | k = 2 D-Acc |
|---|---|---|---|---|---|---|---|---|
| 3.7M | 24 | 27 | 22.0 | 0.017 | 0.0 | 0.0 | 0.0 | 0.0 |
| | 32 | 27 | 24.0 | 0.023 | 0.0 | 0.0 | 0.0 | 0.0 |
| | 32 | 36 | 28.0 | 0.031 | **93.75** | **76.38** | **87.50** | **65.48** |
| 9.8M | 64 | 36 | 28.0 | 0.023 | **95.31** | **70.31** | **84.38** | **60.78** |
| | 96 | 25 | 19.0 | 0.024 | 62.50 | 55.94 | 40.62 | 44.36 |
| | 128 | 22 | 17.0 | 0.029 | 51.56 | 45.09 | 39.84 | 35.92 |

creases, the proportion of ghost sentence tokens in all training tokens rises, making LLMs pay more attention to them. Typically, we use a length around 10 words. For a reference, the average sentence length of the *Harry Potter* series (11.97 words, to be precise) (Haverals & Geybels, 2021). It is worth noting that a long ghost sentences is likely to be filtered by exact substring duplication (Lee et al., 2022), which use a threshold of 50 tokens.

*Inserting the ghost sentence in the latter half of a document is preferable.* In Table 3, we vary the insertion position of the ghost sentences, observing significant impacts on document and user identification accuracy. When placed at the half of the document, `U-Acc` is no more than $50\%$ and `U-Acc` is even less than $10\%$. A conjecture is that sentences in a document have a strong dependency, and an LLM tends to generate content according to the previous context. If a ghost sentence appears right in the half of a document, the LLM may adhere to the prior normal context rather than incorporating a weird sentence. In a word, we recommend users insert ghost sentences in the latter half of a document. Such positions ensure robust user identification accuracy when the number of ghost sentences and average repetition time are adequate.

### 4.3.3 MODEL SIZES AND LEARNING STRATEGIES

*The bigger the model, the larger the learning rate, or the more the epochs, the better the memorization performance.* Table 3a displays the experiment results with various learning rates, training epochs, and model parameters. A larger model exhibits enhanced memorization capacity. It is consistent with the findings of previous works: within a model family, larger models memorize 2-5× more than smaller models (Carlini et al., 2023). This observation implies the potential for commercial LLMs to retain ghost sentences, especially given their substantial size, such as the 175B GPT-3 model (Brown et al., 2020).

The learning rate and training epochs are also crucial. Minor changes can lead to huge impacts on the identification accuracy as illustrated in Table 3a. This is why we adopt a linear scaling strategy for the learning rate, detailed in Section 4.1. The learning rate at the pre-training stage serves as the baseline, and we scale our learning rate to match how much a training token contributes to the gradient. Besides, more training epochs contribute to improved memorization. When a LLaMA-3B model is trained for 2 epochs, it can achieve $100\%$ user identification accuracy. For reference, the training epochs of LLaMA (Touvron et al., 2023a) and GPT-3 (Brown et al., 2020) is $0.64 \sim 2.45$ and $0.44 \sim 3.4$, respectively. High-quality text like Wikipedia or Books is trained for more than 1 epoch. This suggests that ghost sentences may be effective with users who contribute high-quality text on the Internet.

### 4.3.4 CONTINUAL PRE-TRAINING

Previously, we have conducted instruction-tuning experiments to assess the memorization capacity of fine-tuned LLMs for ghost sentences. Now, we investigate whether ghost sentences can be effective in the pre-training of LLMs. However, the pre-training cost is formidable. Training of a "*tiny*" TinyLlama-1.1B (Zhang et al., 2024) model with ~3T tokens on 16 NVIDIA A100 40G GPUs cost

Table 4: **Alignment, wordlist, and data domains**. **(a)** Alignment with DPO. **(b)** OpenLLaMA-3B. **#Words** represents the number of words in the wordlist. $m = 256, \mu = 17$, median $= 13.5$.

(a) **Alignment of TinyLlama-1.1B**.

| #Docs | $m$ | $\mu$ | $k = 1$ | | $k = 2$ | |
|---|---|---|---|---|---|---|
| | | | U-Acc | D-Acc | U-Acc | D-Acc |
| 9.8M | 64 | 36 | **95.31** | **70.31** | **84.38** | **60.78** |
| After Alignment | | | 95.31 | 69.61 | 84.38 | 60.65 |

(b) **Wordlists and training data domains**.

| Domain | Wordlist | #Words | $k = 1$ | | $k = 2$ | |
|---|---|---|---|---|---|---|
| | | | U-Acc | D-Acc | U-Acc | D-Acc |
| Reddit | *Harry Potter* | 4,000 | 77.73 | 76.33 | 66.02 | 68.26 |
| | *Game of Thrones* | 4,000 | 69.14 | 70.02 | 54.69 | 59.36 |
| | *EFF Large* | 7,776 | 92.58 | 91.01 | 84.77 | 84.66 |
| | *Natural Language* | 7,776 | 88.28 | 87.67 | 78.52 | 78.27 |
| | *Niceware* | 65,536 | 94.92 | 94.96 | 91.02 | 89.63 |
| Patient Conv. Code | *EFF Large* | 7,776 | 77.73 | 79.22 | 62.11 | 67.49 |
| | | | 99.22 | 99.10 | 98.44 | 98.74 |

90 days. Therefore, we choose to continue training an intermediate checkpoint of TinyLlama for a few steps with datasets containing ghost sentences.

*Larger repetition times of ghost sentences are required for a "tiny" 1.1B model and millions of examples.* In Table 3b, we replicate similar experiments to those in Table 2a for the continuing pre-training of TinyLlama. To make a 1.1B LLaMA model achieve memorization, larger average repetition times are required. This is consistent with Table 3a, where a 1.1B LLaMA model cannot remember any ghost sentences. By contrast, 3B and 7B LLaMA models achieve good memorization. To better understand this point, we provide visualization of D-Acc-1 with different $\mu$ for TinyLlama in Figure 7 in §App. F.

### 4.3.5 ALIGNMENT, WORDLIST, AND DATA DOMAIN

*Limited steps of alignment will not affect the memorization of ghost sentences.* After pre-training and fine-tuning, modern LLMs will be further aligned for helpfulness, honesty, and harmless (Bai et al., 2022; Ouyang et al., 2022). Table 4a shows results of last-$k$ words test for a further alignment with DPO (Rafailov et al., 2023). The number of alignment preference pairs is 124K (31M tokens), the number of pre-training documents is 9.8M, and the proportion of preference tokens is 0.0123%. For reference, LLaMA-2 (Touvron et al., 2023b) uses 2.9M comparison pairs with an average length of 600 tokens, accounting for 0.00087% of the 2T pre-training tokens.

*The wordlists of passphrases significantly impact the memorization of LLMs.* In the above experiments, we use diceware passphrases generated from the EFF Large Wordlist published by the Electronic Frontier Foundation (EFF). Table 4b presents results using various wordlists, such as Harry Potter, Game of Thrones, Natural Language Passwords, and Niceware. Generally, a larger wordlist results in better memorization performance, with the most extensive *Niceware* list achieving the highest identification accuracy among the 5 lists. Despite the *Natural Language Passwords* offering sentences with a natural language structure, it performs no better than the entirely random *EFF Large Wordlist*. Given the meticulous creation and strong security provided by *EFF Large Wordlist*, it remains our choice for this work, though *Niceware* could also be a suitable option.

*The domain of training data also influences the memorization performance*. Table 4b showcases experiments conducted with 100K real patient-doctor conversations from HealthCareMagic.com (Li et al., 2023) and 120K code examples (iamtarun/code_instructions_120k_alpaca). Ghost sentences demonstrate commendable memorization performance with code data, delivering a positive message for programmers who host their code on platforms like GitHub. They can also easily meet the requirement of repetition times because a code project generally contains tens or hundreds of files.

## 5 CONCLUSION

In this work, we propose an *insert-and-detection* methodology for membership inference of online copyrighted material. Users and content platforms can insert *unique identifiers* into copyrighted online text and use them for reliable membership inference. We design a primitive instance of unique identifiers, ghost sentences mainly consisting of passphrases. Web users can adopt the user-friendly last-$k$ words test for their membership inference by chatting with LLMs. Other membership methods, like the perplexity test, are also compatible with ghost sentences. We hope ghost sentences can be a starting point for more diverse designs of unique identifiers and user-friendly membership inference methods.

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

## A  BROADER IMPACTS

The proposed unique identifiers assist web users in protecting online copyright material in large language model training. Ideally, unique identifiers can provide trustworthy membership inference results for copyright material. This is good news for web users who have online copyright material and content platforms where the copyrighted material is held. Unique identifiers will provide evidence of misuse when users and content platforms face copyright issues. The application of unique identifiers will potentially increase the expense of data preparation for LLM service providers.

## B  RELATED WORKS

**Instruction Tuning**   The most popular fine-tuning method for pre-trained LLMs now is instruction tuning (Wang et al., 2023). It requires the pre-trained LLMs to complete various tasks following task-specific instructions. Instruction tuning can improve the instruction-following capabilities of pre-trained LLMs and their performance on various downstream tasks (Taori et al., 2023; Chiang et al., 2023; Mukherjee et al., 2023; Li et al., 2023; Xu et al., 2023). The training data for instruction tuning come from either the content generated by powerful commercial LLMs like GPT-4 (Taori et al., 2023; Mukherjee et al., 2023), or data from web users (Chiang et al., 2023; Li et al., 2023).

**Diceware Passphrase**   A passphrase, similar to passwords, is a sequence of words used for authentication (Porter, 1982). Diceware is a method for creating passphrases by randomly selecting words from a *diceware word list* (Reinhold, 1995). This list typically consists of $6^5 = 7776$ words (determined by rolling dice five times). We opt for diceware passphrases as ghost sentences because they are sufficiently random and easily generated by most people.

## C  VERBATIM MEMORIZATION CAPABILITY OF COMMERCIAL LLMS

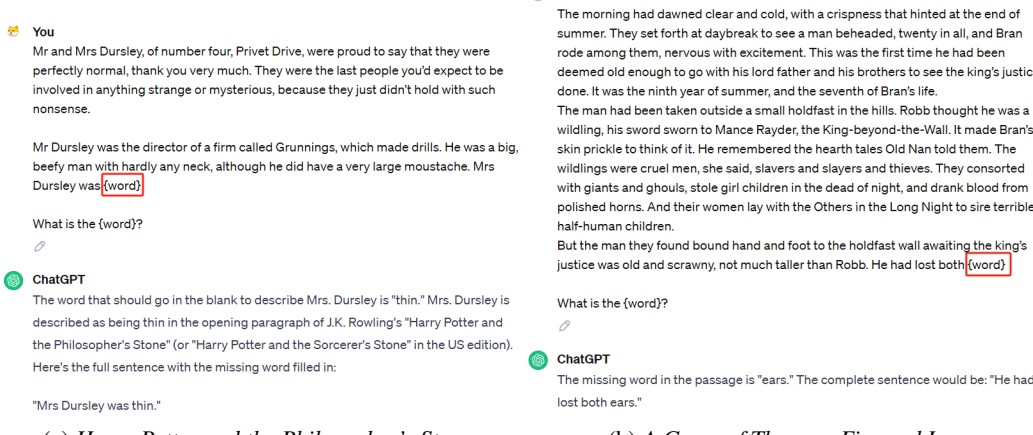

(a) *Harry Potter and the Philosopher's Stone.*          (b) *A Game of Thrones, Fire and Ice.*

Figure 5: **ChatGPT can achieve verbatim memorization for popular books.** ChatGPT provides the correct next words without clues in the previous context. Conversations happen on 18/01/2024 with ChatGPT-3.5. Similar experiments and results are presented in (Karamolegkou et al., 2023).

Commercial LLMs like ChatGPT can memorize the content of popular books verbatim as shown in Figure 5. Some conclusions can be drawn from the phenomenon: 1) This demonstrates the significant memorization capacity of LLMs. 2) OpenAI may not have a strict process for deduplicating repeated content in the training data. Otherwise, verbatim memorization would not be possible. It is also possible that a strict deduplication process could lead to worse performance of LLMs, especially for short pieces of text, as this could break the integrity of the whole text.

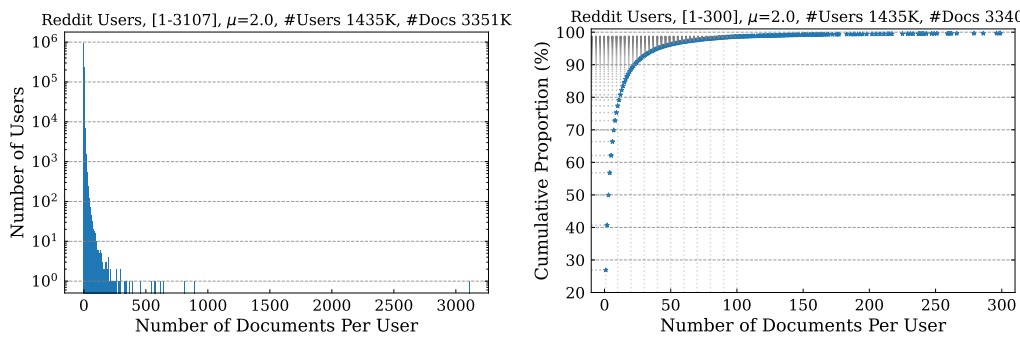

(a) **The number of documents per user.**     (b) **The cumulative document proportion**.

Figure 6: **Statistic of Reddit user data (Völske et al., 2017)**. **(a)** The y-axis is logarithmic. $\mu$ represents the average number of documents per user. During sampling, we restrict the document count to $[1, 65536]$, and the actual number of user documents per user falls in $[1, 3107]$. A special user [deleted] has 374K documents. It is a system user, and we ignore it. **(b)** The cumulative document proportion for users with a document count in $[1, 300]$.

Table 5: **Results on HellaSwag and MMLU. #Docs** is the number of training examples, **mid.** is the median of repetition times, and **prop.**(%) is the proportion of documents with ghost sentences in all examples. The length of ghost sentences is 12. U-Acc and D-Acc refer to Table 2a.

| #Docs | $m$ | $\mu$ | mid. | prop.(%) | HellaSwag | MMLU |
|---|---|---|---|---|---|---|
| OpenLLaMA-3Bv2 Geng & Liu (2023) | | | | | 69.97 | 26.45 |
| | 256 | 17 | 13.5 | 2.99 | 71.23 | 26.01 |
| | 128 | 17 | 13.0 | 1.47 | 71.32 | 26.10 |
| | 64 | 17 | 13.0 | 0.74 | **71.46** | 26.13 |
| | 32 | 18 | 12.0 | 0.39 | 71.39 | 26.36 |
| | 16 | **13** | **11.5** | 0.14 | 71.43 | 25.85 |
| 148K | 16 | 21 | 16.5 | 0.22 | 70.94 | 25.40 |
| | 8 | 18 | 13.0 | 0.10 | 71.35 | 26.29 |
| | 8 | 31 | 25.5 | 0.16 | 71.32 | 25.38 |
| | 4 | 32 | 32.5 | 0.087 | 71.00 | 25.96 |
| | 2 | 48 | 47.5 | 0.064 | 70.88 | 25.74 |
| | 1 | 45 | 45.0 | **0.030** | 70.39 | 25.37 |
| | 1 | 51 | 51.0 | **0.034** | 70.40 | 25.37 |
| 148K | | | | 0.259 | 70.55 | 26.21 |
| 592K | 16 | 24 | 20.5 | 0.065 | 70.76 | **26.64** |
| 1.8M | | | **0.022** | | 71.07 | 26.51 |

# D    STATISTICS OF USERS ON REDDIT

Figure 6 displays the statistics of users in Webis-TLDR-17 (Völske et al., 2017), which contains Reddit subreddits posts (submissions & comments) containing "TL;DR" from 2006 to 2016. Figure 6a shows that the number of documents per user mainly falls within the range of $[1, 300]$, with a long tail distribution. This is evident in Figure 6b. Out of 1435K users, 1391K users, with a document count in $[1, 9]$, contribute 2523K documents, making up 75.3% of the total 3351K data.

# E    RESULTS ON COMMON BENCHMARKS

In Table 5, we provide the results for instruction tuning on common benchmarks like HellaSwag (Zellers et al., 2019) and MMLU (Hendrycks et al., 2021). Table 5 corresponds to identification results in Table 2a. Table 5 shows that inserting ghost sentences into training datasets has no big influence on the performance of LLMs on common benchmarks.

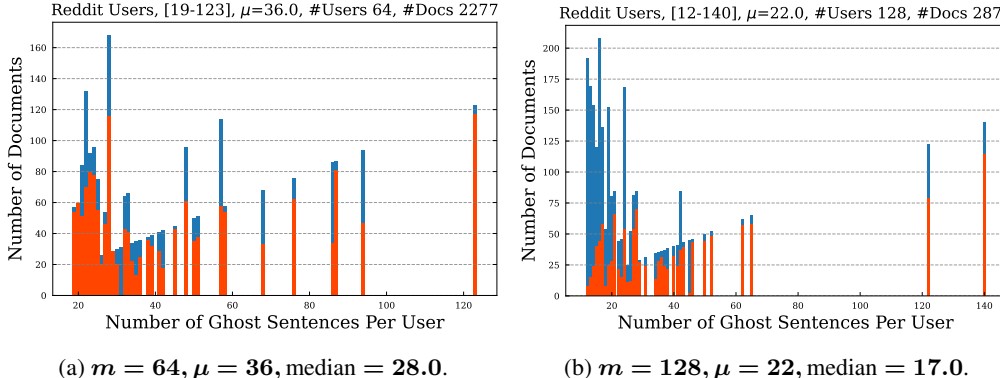

(a) $m = 64, \mu = 36$, median $= 28.0$.    (b) $m = 128, \mu = 22$, median $= 17.0$.

Figure 7: **D-Acc-1 of TinyLlama with various repetition times**. The **blue bar** defines the number of documents, and the **orange bar** represents the number of ghost sentences that are correctly memorized by the LLM. The total training data is 9.8M.

Table 6: **Last $k = 2$ words generation for ghost sentences with a TinyLlama-1.1B model**. Ghost sentences are in blue. These two cases come from a user W★★★a who has 38 examples in Webis-TLDR-17 Völske et al. (2017).

---

**prompt**: ...[**omit**]... Instruction:\nContinue writing the given content\n\n### Input:\nHad an appendicitis the night before a flight to Hawaii.\n\n### Response:\nDocs said that had it happened mid-air, I might have died since there are few good options for diverting a plane for a med emergency over the Pacific. Ruined my vacation and started my pregnant wife on the road to hating me and wanting a divorce, but I lived. headache exterior subtext semifinal rewrap rupture animal aim wrath splendor
**Generation**: rash blaspheme
**Ground Truth**: rash blaspheme

---

**prompt**:...[**omit**]...Input:\nHere goes: \nAuthors like Tolkien, ...[**omit 122 words**]...\n\n### Response:\nIn Martin's works however, we are dealing with huge swaths of territory where everyone magically communicates in the same language, ...[**omit 143 words**]... headache exterior subtext semifinal rewrap rupture animal aim wrath splendor
**Generation**: rash blaspheme
**Ground Truth**: rash blaspheme

---

## F  IDENTIFICATION ACCURACY OF TINYLLAMA

Figure 7 presents the D-Acc-1 with different repeat times of ghost sentences for TinyLlama. Compared to larger LLaMA-3B or LLaMA-7B models (Table 3a), the "tiny" 1.1B models require larger repetition times of ghost sentences.

In Table 6, we give two examples to show how we generate the last $k = 2$ words of ghost sentences with TinyLlama.

## G  EXAMPLES WITH GHOST SENTENCES

We show three examples with ghost sentences here. The first one is from HealthCareMagic.com (Li et al., 2023), and the other two examples come from a Reddit user S★★★hy, who have 86 exmplaes in Webis-TLDR-17 dataset (Völske et al., 2017). Ghost sentences are in blue.

⌂ *Example 1*:
**Instruction**: If you are a doctor, please answer the medical questions based on the patient's description.
**Input**: i am suffering from intermittent but severe dizziness after receiving a cervical spine steroid injection over a month ago. ...[**omit 36 words**]...

**Output**: Hello, The blood vessels that supply the inner ears and the parts of the brain that are connected with the balance of the body pass through the cervical vertebrae. ...[**omit 54 words**]... Let me know if I can assist you further. Take care Chat Doctor. ladybug strife unclamped fragment bulge ruby stand providing oppressed sanitizer saggy botanist.

👻 *Example 2*:
**Instruction**: Continue writing the given content
**Input**: So, thanks for the clarification regarding ARS. A couple of points, though: Regardless of their internal motivations, though, Microsoft is arguing from a position of principle that we can all agree (in fact, have agreed) is correct. Not on the page you linked to. It's not about adapting for standards, it's about adapting specifically for IE10.,
**Output**: For example, the blog says this: ...[**omit 38 words**]... It's not an argument at all, or any sort of political statement. It's not really good, or bad. ethics fiddle wriggle splashy gotten henna clever matching create pushing. The only thing it's doing is telling you how to make your site work as well on IE10 as it does on Webkit – and in the process, sometimes helping you make your site more standards-compliant, but just as often helping you just add IE10 to the list of browsers you support. I'm also not sure how you get to your

👻 *Example 3*:
**Instruction**: Continue writing the given content
**Input**: Do you guys think my deception early in the relationship was wrong? Yes. Am I wrong to have married a Christian? ...[**omit 86 words**]...
**Output**: Relationships have survived far worse than this. ...[**omit 166 words**]... I would be surprised if this was an issue, though, given how long you've been together. ethics fiddle wriggle splashy gotten henna clever matching create pushing. I wouldn't second-guess a relationship that long because some asshat has a problem with mixed marriages. Anyway

## H   THE GHOST SENTENCE OF THIS WORK

This paper contains 12 mysterious words: quickstep drudge consent wackiness mangle unspoiled childish exploring antennae agony embassy starved.

