# OpenReview forum: "Protecting Copyrighted Material with Unique Identifiers in Large Language Model Training"
_ICLR.cc/2025/Conference — ICLR 2025 Conference Withdrawn Submission_

### Official Review · Reviewer_HCgh · 2024-10-29

**Soundness:** 3
**Presentation:** 2
**Contribution:** 2
**Rating:** 3
**Confidence:** 4

**Summary:**

The paper addresses the problem of membership inference in LLMs, where the goal is to determine whether a particular document or set of documents was used for training. The authors motivate this with concerns about the use of copyrighted material. They propose the insertion of _ghost sentences_ — sentences made up of randomly sampled words — into online documents. To test whether a model has been trained on documents containing a specific ghost sentence, they test two methods: 1. Computing the perplexity of the ghost sentence under the model probability distribution compared with the perplexity of other, randomly sampled sentences. 2. Prompting the model to guess words in the ghost sentence. A significantly lower perplexity (in case 1) or a successful guess (in case 2) indicates the presence of the ghost sentence in the training data. The authors perform various experiments covering both pretraining and fine-tuning settings, different model size, ghost sentence lengths and other parameters. They show that, given enough ghost sentences, their method can successfully detect the presence of ghost sentences in the training data of LLMs.

**Strengths:**

- The authors motivate the last-k word test by the need for easy-to-use membership inference methods that can be deployed by average users. I agree that this is an important and often overlooked aspect that can stand in the way of a broad adoption of such methods.
- Besides the ease of use, the method also comes with the advantage of not relying on output probabilities that are often not available for commercial models.
- The authors do a good job at covering various settings in their experiments. I particularly appreciate that they perform an experiment to determine whether ghost sentences inserted during pretraining are still detectable after alignment training.
- The introduction serves as a very good overview and summary of the paper.

**Weaknesses:**

- My main criticism of the paper is the lack of novelty. The authors cite Wei et al. [1], who propose a very similar approach of introducing random sequences of characters (instead of random sequences of words) into documents. They mention three disadvantages of [1] when compared to their own method: 1. The model trainer could filter out the random character sequences. 2. There is a risk of false positives due to the prevalence of such random sequences in training data. 3. Random character sequences do not allow for a user-friendly detection method. I think that criticism only holds to some degree: 1. The model trainer could filter out random sequences of words using a method similar to the proposed perplexity test applied to a previously trained, small language model. This would arguably be harder than simply filtering out very uncommon sequences of characters, but still seems feasible. 2. If the sequence of random characters is sufficiently long, this is not an issue, because a collision becomes very unlikely. 3. It seems possible to me that the last-k prompt could be used with random sequences of characters, where instead of asking the model to complete a sentence, once asks it to complete a sequence of characters.
- I would have liked a comparison with [1] in terms of detection success, and an experiment confirming or refuting point 3. above.
- Taken independently, neither the perplexity-based hypothesis test is new [2] nor the idea of checking whether the model can determine a correct word in a sentence better than random guessing [3,4].
- The instruction tuning task “Continue writing the given content” is very similar to the pretraining task and not a task one would expect during instruction tuning.
- It seems like when applying the method to pretraining data, an unrealistically large percentage of the data needs to be ghost sentences (cf. Table 3b), given that modern LLMs are trained on trillions of tokens. However, that is the training phase where most data is used and thus the most relevant one for detecting the use of copyrighted material. It could in principle be that the required percentage decreases to manageable levels for large models that have been shown to memorize more data than small models. To avoid the cost of (continuous) pretraining of such models, one way to test that hypothesis might be to follow an approach similar to that of [1] and try to find instances of random word sequences in the training data of large models with publicly available training sets such as BLOOM.
- Minor: The sentence starting in 185 is not grammatical. And the first half of Sec. 4 has worse writing than the rest of the paper. Besides several typos and grammar mistakes, the beginning of the paragraph in ll. 324-329 ("To figure out the average repetition $\mu$ [...]") does not seem to seem to make sense in the context of the rest of the paragraph.
- Minor: Throughout the paper, the authors use the term 'word length' several times, which I would interpret as the number of characters in a word. It seems like what is rather meant is the number of words in a ghost sentence. If that is the case, I suggest to change the term to 'sentence length'.


[1] Wei, Johnny Tian-Zheng et al. Proving membership in LLM pretraining data via data watermarks. arXiv preprint arXiv:2402.10892, 2024.

[2] Carlini, Nicholas et al. Membership inference attacks from first principles. 2022 IEEE Symposium on Security and Privacy.

[3] Lukas, Nils et al. Analyzing leakage of personally identifiable information in language models. 2023 IEEE Symposium on Security and Privacy.

[4] Chang, Kent K. et al. Speak, memory: An archaeology of books known to ChatGPT/GPT-4. arXiv preprint arXiv:2305.00118, 2023.

**Questions:**

- 195-196: As far as I can see, we can always upper bound $1/V^*$ by $1/V_g$. Seeing ghost sentences other than the one we are currently interested in gives the model information about the set of words from which the ghost sentences are created, but no information beyond that. Thus, can't we simply plug $V_g$ into Eq. 3? Since we can choose $V_g$ arbitrarily large, we do not need to hope that we get $n_g=2$. Could you clarify?
- Why do you use beam search for decoding instead of sampling-based decoding methods that are more commonly used for general-purpose LLMs? How would using such a decoding method change your results?

---

### Official Review · Reviewer_YYzf · 2024-10-30

**Soundness:** 3
**Presentation:** 3
**Contribution:** 3
**Rating:** 6
**Confidence:** 3

**Summary:**

This paper presents an insert-and-detect method to detect whether a text is included in LLM training data. The proposed method is based on unique identifiers.

**Strengths:**

1. Detect training data in LLMs is important for copyright protection.

2. The proposed method is effective in detecting training data.

3. The experiments are extensive and solid.

**Weaknesses:**

1. Can the perplexity test be applied to close-source models?

2. This paper uses finetuning to inject the unique identifiers. However, the problem studied in this paper is for detecting training data in pretraining, right? Is it possible that the finding and conclusion obtained from finetuning experiments be different for pretraining?

3. What are existing data filtering methods for LLM pretraining data preprocessing? Is the proposed method robust to these data filtering methods?

4. Is it possible the proposed method can harm the quality and readability of the data to be protected?

**Questions:**

1. Can the perplexity test be applied to close-source models?

2. This paper uses finetuning to inject the unique identifiers. However, the problem studied in this paper is for detecting training data in pretraining, right? Is it possible that the finding and conclusion obtained from finetuning experiments be different for pretraining?

3. What are existing data filtering methods for LLM pretraining data preprocessing? Is the proposed method robust to these data filtering methods?

4. Is it possible the proposed method can harm the quality and readability of the data to be protected?

---

### Official Review · Reviewer_TMjr · 2024-11-01

**Soundness:** 2
**Presentation:** 2
**Contribution:** 2
**Rating:** 5
**Confidence:** 5

**Summary:**

The paper introduces a novel method for protecting copyrighted content in language models by embedding unique identifiers, termed "ghost sentences," into text data. This technique enables content owners to identify unauthorized use of their data by querying LLMs. The proposed framework includes the "last-k words" and "perplexity" tests for efficient, accessible membership inference.

**Strengths:**

- Presents an innovative approach to copyright detection using ghost sentences, filling a niche not fully addressed in LLM training.
- Provides extensive experimental validation with different model sizes, data scales, and insertion strategies.

**Weaknesses:**

- Writing needs improvement, including but not limited to introduction
- The example in Figure 1 is not self-explanatory
- notations are confusing. Section 3.1 an example is indicated by both $x_i$ (with subscript) and $x$ (without subscript).
- Dependence on specific models and configurations, which might not generalize well across all LLM architectures.
- Experiments were only performed on the LLaMA family. There are alternative open-source models such as Mistral.
- If I understand correctly, the model should also work on proprietary models. However, experiments on state-of-the-art proprietary models such as GPT-4o are missing (see https://openai.com/index/gpt-4o-fine-tuning/)
- Line 73 "Figure Figure 1"

**Questions:**

- Does training with ghost sentence impact generative fluency (lower the models' abilities to generate fluent sentences)?

---

### Official Review · Reviewer_9zEV · 2024-11-03

**Soundness:** 3
**Presentation:** 3
**Contribution:** 2
**Rating:** 3
**Confidence:** 4

**Summary:**

The paper proposes using "ghost sentences"—unique, embedded passphrases—to help creators detect if their copyrighted content has been used in LLM training, offering methods like the last-k words and perplexity tests for reliable, user-friendly detection.

**Strengths:**

1. The proposed method introduces accessible detection techniques, allowing non-technical users to check if their content has been used for LLM training.
2. The study thoroughly examines factors like model size, training data scale, and repetition frequency, ensuring the robustness of the proposed approach across different LLM configurations.

**Weaknesses:**

1. Perplexity-based filtering is a common method for cleaning pre-training data. How might the proposed ghost sentences method be adapted to remain effective in the presence of perplexity-based data cleaning techniques commonly used in LLM training?
2. Could the authors provide a more detailed analysis of how different wordlist impact the effectiveness of ghost sentences? Additionally, what are the trade-offs of using the entire LLM vocabulary as the wordlist versus a smaller, curated list?
3. With LLM-generated or rephrased training data becoming more popular, it would be interesting to investigate whether ghost sentences persist after LLM-based rewriting.
4. The paper lacks baseline comparisons to validate its effectiveness.

**Questions:**

N.A.

---

### Official Review · Reviewer_xPDZ · 2024-11-03

**Soundness:** 3
**Presentation:** 3
**Contribution:** 2
**Rating:** 5
**Confidence:** 4

**Summary:**

The paper proposes a method to detect the presence of copyrighted material of a user in the training corpus of LLMs. The authors suggest a method that uses unique, randomly generated passphrases as identifiers, which are to be embedded into the user's text data. To detect these passphrases within the model's outputs, the study suggests two main techniques: a "last-k words" test and a perplexity-based test. The paper argues that these detection methods are accessible and user-friendly. They also conduct ablations analyzing several aspects such as scale of training data, model sizes, repetition of passphrases, etc.

**Strengths:**

This is definitely an important problem which is clearly motivated. I appreciated that the authors conducted detailed ablations analyzing the memorization of the ghost sentences and presented detailed results with repetition rates, model sizes, length and insertion position, etc. The suggested techniques are also simple and intuitive.

I also liked that the paper proposes a method to get interpretable p values for both the user-friendly k-words test and the perplexity based text.

**Weaknesses:**

The paper has the following limitations:

- In my view, the work lacks novelty considering the overlap with [1] and [2] which introduce multiple varieties of data watermarks (or copyright traps) into the text and detect membership using loss-based metrics. The more practical perplexity-based test is also similar to the one proposed in [1] which compares the loss-based metric of a watermark against the empirical distribution. I believe it would help to include a detailed comparison with [1] and [2] by highlighting significant benefits of the passphrases proposed in the paper.

- Another limitation is that the more aggressive perplexity-based text requires the watermark to be repeated a significant number of times (5 times) and still offers a recall of just ~0.393.


1. [Proving membership in LLM pre training data via data watermarks](https://arxiv.org/abs/2402.10892) by Wei et al.

2. [Copyright Traps for Large Language Models](https://arxiv.org/abs/2402.09363) by Meeus et al.

**Questions:**

- The paper suggests that the watermarks in [1] run the risk of being filtered? Could you provide evidence of a practice that data providers use to filter this?

- The paper offers details on the fine-tuning in terms of number of examples used, could you please share the details in terms of token counts?

- What would be the result of the baseline provided for the min-k test using the best performing hyperparameters (k value)?

1. [Proving membership in LLM pre training data via data watermarks](https://arxiv.org/abs/2402.10892) by Wei et al.

---

### Note · Authors · 2024-12-02

I have read and agree with the venue's withdrawal policy on behalf of myself and my co-authors.